# Analysis of the Machining Process of Titanium Ti6Al-4V Parts Manufactured by Wire Arc Additive Manufacturing (WAAM)

**DOI:** 10.3390/ma13030766

**Published:** 2020-02-07

**Authors:** Fernando Veiga, Alain Gil Del Val, Alfredo Suárez, Unai Alonso

**Affiliations:** 1TECNALIA, Basque Research and Technology Alliance (BRTA), Parque Científico y Tecnológico de Gipuzkoa, E20009 Donostia-San Sebastián, Spain; alain.gil@tecnalia.com (A.G.D.V.); alfredo.suarez@tecnalia.com (A.S.); 2Department of Mechanical Engineering, University of the Basque Country (UPV/EHU), E48013 Bilbao, Spain; unai.alonso@ehu.eus

**Keywords:** additive manufacturing, hybrid manufacturing, WAAM, PAW

## Abstract

In the current days, the new range of machine tools allows the production of titanium alloy parts for the aeronautical sector through additive technologies. The quality of the materials produced is being studied extensively by the research community. This new manufacturing paradigm also opens important challenges such as the definition and analysis of the optimal strategies for finishing-oriented machining in this type of part. Researchers in both materials and manufacturing processes are making numerous advances in this field. This article discusses the analysis of the production and subsequent machining in the quality of TI6Al4V produced by Wire Arc Additive Manufacturing (WAAM), more specifically Plasma Arc Welding (PAW). The promising results observed make it a viable alternative to traditional manufacturing methods.

## 1. Introduction

Wire Arc Additive Manufacturing (WAAM) is an additive manufacturing (AM) technology that uses a metal wire and a welding by arc as an energy source. In this process, the AM head melts the tip/point of the metal wire, creating a high-quality metallurgical union between the substrate and the added material. The material deposition is carried out layer by layer, and it finishes when the geometry is developed [1]. Currently, titanium and nickel alloys are among the most investigated materials used in WAAM technology [2].

The main benefit of WAAM, as compared to other technologies such as Selective Laser Melting (SLM) or Laser Metal Deposition (LMD), is its high deposition rate (3–8 kg/h). Other advantages are its cost-competitiveness and the environmental friendliness [3,4]. Moreover, using the material in a wire form is a competitive benefit respect of metal powder technologies because it has already been validated in the welding industry. Therefore, WAAM is the best solution to manufacture machine parts with medium complexity, medium–large sizes, and high mechanical properties in aeronautical, oil and gas, and railway sectors [4]. However, the commercialization of such parts in the aircraft industry requires more studies to validate its feasibility [5].

Regarding production costs, the buy-to-fly ration is very small in alloys such as Ti6Al-4V. These materials are manufactured by a forging process followed by a machining operation that gives its final geometry. To solve this, WAAM seems to be a high-material deposition technology with this material and, contently, reduces the machining time and costs [1,6].

WAAM is a near net shape technique because it does not obtain the final profile [1]. Besides, the surface finish and dimensional accuracy are poor and low, respectively. Due to the geometrical and mechanical requirements of the aerospace sector, machining processes are needed to improve the smoothness of the surface and the required dimensional accuracy. Therefore, the most common approach is to finish the additively manufactured parts by an end-machining process [1].

In this regard, the concept of hybrid manufacturing could be feasible by the combination of WAAM and subtractive processes. Furthermore, this manufacturing solution includes the benefits of the AM techniques and guarantees the final machining requirements [7,8], but there are some downsides on its industrialization [9].

In the past, several investigations have studied the effect of the mechanical and chemical properties of workpieces produced by AM on residual stresses and chip formation [10,11]. Some investigators have integrated AM technology in one CNC (Computer Numerical Control) machine [12]. Currently, an assessment illustrates the challenges and the addressing of a hybrid manufacturing future [13]. Cranfield University is a pioneer in this topic, particularly in WAAM and machining processes. Qiao et al. [14] manufactured steel and stainless-steel multilayer walls optimizing the WAAM process parameters. They studied geometry accuracy, stress concentration, and the stability of deposition material when milling a WAAM manufactured wall. Furthermore, they are working on the LASSIM European project, whose objective is to develop a high productive hybrid manufacturing machine for large parts [15]. Montevechi et al. [16] compared the magnitude of milling forces in workpieces manufactured by laser deposition (LENS) and WAAM technologies. Prado-Cerqueira et al. [17] developed a hybridation in a CNC machine. In this work, the authors manufactured WAAM components using CNC code.

The material usage of titanium alloys has soared 9% in the aircraft sector; particularly from Boeing 747 to 787 aircraft [15]. This high increment is due to its outstanding mechanical characteristics, its corrosion resistance, and its low density. However, the machining is demanding due to high-level cutting forces and torque [18]. In this sense, mechanical and machining properties were analyzed in the past for workpieces produced by different AM technologies [4,19,20].

The aim of this paper is to investigate the influence of the PAW-WAAM technique of Ti6Al-4V alloy on the subsequent milling operation. Firstly, the manufacturing process (additive on PAW-WAAM plus milling operation) is described together with the signal-acquisition system, the cutting experiments, and the workpiece quality analysis procedure. Secondly, mechanical and microstructural results are shown. Besides, the results of the cutting forces, the torque, and the bending moments are discussed, and surface quality results are also presented.

## 2. Materials and Methods

### 2.1. Experimental Setup for Hybrid Additive/Subtractive Manufacturing

Plasma Arc Welding (PAW) is the selected WAAM technique. This high energy density process manufactures high-quality walls with high deposition rates and without showing the metallurgical problems or arc welding, such as the lack of fusion and porosity. Moreover, PAW presents fewer problems of distortion and heating. For titanium and stainless steel, this technique leads to high mechanical specifications for components of medium–large size [21].

Figure 1 shows the 5-axis Addilan WAAM machine [22] with a PAW welding system. The open CNC software together with its particular architecture gives the machine a high versatility. Moreover, it also enables the production of parts using some additional AM technology such as Gas Tungsten Arc Welding (GTAW) and Gas Metal Arc Welding (GMAW).

One of the main features of this equipment is its specific close chamber to work with reactive materials such as titanium or aluminum. Prior to additive manufacturing (AM), the chamber is filled with argon to protect the melt pool of oxidation and porosity. This machine was used to manufacture the walls of Ti6Al4V under an oxygen level of 50 ppm using argon as the protection gas. The wall was separated into a set of two samples machined on a CNC machining center: one to characterize the mechanical properties and the other one to perform the machining tests. For that purpose, a titanium substrate of 12 mm thickness was employed and 1.2 mm wire diameter (AWS A5.16-13 ERTi-5 Titanium Welding Wire that meet standard AMS-4954J). Thus, the deposition rate is 2 kg/h. Table 1 shows the chemical composition of the wire material.

The manufactured Ti6Al4V wall was developed by two overlapped beads at 50%, and the inter-bead temperature is 600 °C. Finally, a post heating treatment was performed at 720 °C on the wall along 2.5 h to characterize the welding material.

### 2.2. Machining Strategy and Cutting Tool

Figure 2a illustrates the measurements carried-out in a 5-axis ZV 65/U600 EXTREME vertical CNC tool machine belonging to the Ibarmia manufacturer. During the machining process, the Pro-micron Spike^®^ system was used to acquire the cutting force in the Z direction, the torque, and the bending moments. This sensor is located on the tool holder and collects the tool-bending moments along the X and Y directions (Figure 2b). The post-processing analysis of registered signals was performed by Python^®^ software.

The tool diameter and teeth number of the KENDU milling tool are 12 mm and 5 teeth, respectively. The milling tool, which can be seen in Figure 2b, is made of WC-8%Co material. Before describing the machining measurements, a walls preparation is carried out to establish a constant axial and radial depth of cuts, as can be noticed in Figure 3.

Figure 3a shows the poor surface finish and dimensional accuracy on the wrought wall after the PAW-WAAM process. In order to study the effect of the workpiece material on the process (AM versus conventional titanium alloy), a uniform radial depth of cut should be obtained during the milling operation. To do so, the workpiece manufactured by AM was pre-machined. Figure 3b illustrates the machined wall after its preparation. This previous machining operation is also mandatory for the study of the material properties.

Before explaining the tests to be performed, Figure 4 shows the three milling strategies studied in the PAW-WAAM workpiece. The first one is an up-milling strategy (Figure 4a), and the cutting tool is fed in the direction of rotation. The second milling strategy (Figure 4b) is a down-milling operation (also called climb milling) in which the workpiece is fed against the direction of the cutter rotation. The last one (Figure 4c) is slot-milling operation. This is an end-milling operation in which the side and face milling are available.

The deposition path used in the PAW-WAAM operation has been overlapped in Figure 4a. This path was the same for all the strategies and could influence the surface finish or dimensional accuracy when machining with a different strategy. Therefore, three sets of experiments were defined to analyze the correlation among cutting parameters, milling processes, and PAW-WAAM deposition rate.

In the first set of experiments, different milling strategies were compared, and the effect of the feed was analyzed. As shown in Table 2, the axial (ap) and radial (ae) depths of cut and the cutting speed (Vc) were set constant, and the feed (fz) and milling strategies were varied. It should be noted that the same table configuration has been used to illustrate the machining conditions: milling strategy in the first column and machining parameters in the next ones. In the second set of experiments (Table 3), the influence of the tool depth in the Z axis was evaluated. This parameter is the product of the depth of cut and the number of passes (starting from the top of the wall). Finally, the effect of the cutting speed was studied in the third set of experiments (Table 4). For these tests, an up-milling strategy was chosen.

To sum up the measurement configuration, the aim of this experimentation is to study the influence of milling strategies and feed, the Z depth, and the cutting velocity on the torque and surface quality of the final PAW-WAAM part.

### 2.3. Workpiece Quality Analysis

The mechanical properties of the PAW-WAAM components were determined by tensile testing to check the possible variation of the mechanical properties according to ISO 6892-1 standards because PAW-WAAM is a directional manufacturing technique (vertical and horizontal). Besides, a metallic microstructure analysis has been performed. Samples were sectioned along the XZ, XY, and YZ planes of the workpiece. Then, a polishing and etching in Kroll’s were performed to analyze the macro and microstructure, and optical images of the microstructure were obtained. Figure 5 shows the specimens’ characteristics being evaluated according to ISO-6892-1 and the configuration/orientation of specimens for learning the added material properties.

The ERNST Computest light load (5 kgf) hardness tester using the Rockwell method was used according to ISO 6508 procedures on the wall manufactured by the PAW-WAAM process. Characterization of the machined wall surface roughness was made by using a Leica DCM 3D dual core measuring system.

## 3. Results

### 3.1. Mechanical Properties and Workpiece Microstructure Characterization

In this section the quality of the workpiece is analyzed. The main characteristics of the piece in its pre-machining state: chemical composition, microstructure, microhardness, and mechanical properties have been measured, as described in the previous section.

The following Table 5 shows the chemical composition of the wall once manufactured by means of the PAW-WAAM additive technology. The concentration of the different elements is included among the reference values established by the standard that describes this titanium alloy for aeronautical use from foundry (AMS 4928). There are no concentrations outside the established oxygen, nitrogen, and hydrogen possible, given the high affinity of these elements with titanium alloys at high temperatures, which have been described by other authors in different additive technologies [23].

The microstructure analysis reveals the presence of a columnar grain structure along the XZ direction, as it can be seen in Figure 6. This type of microstructure is common in the titanium alloy produced by additive technology, but it does not correspond to the structures with equiaxial grains, which are typical of the traditional manufacturing processes of this titanium alloy [24].

The results of surface hardness in the wall at different depths are shown in Figure 7. The average hardness observed is slightly higher than 40 HRC. There are no large differences in the hardness values at different depths, although the average hardness in the middle area of the wall is greater, this difference cannot be established as significant due to the high variance observed.

Table 6 shows the results of the tensile test of the specimens extracted from the titanium alloy wall produced under the same PAW-WAAM process conditions. These tensile tests allow us to establish the mechanical behavior of the material produced by this technology that has adequate results of both ultimate tensile strength (UTS), yield stress (YS), and elongation before fracture. Although, the fact that certain values of UTS and yield stress from the tests in the vertical direction have shown values below the limit established by the standard would make a subsequent thermal treatment that will improve the mechanical behavior necessary. These values are among those requested by the aeronautic standard (AMS4928) that applies, as it can be seen in Table 6. Finally, it should be noted that it is seen in Figure 8 that the results are slightly worse when the specimens are extended in the vertical direction, since they are located between the different strands that make up the wall. These results are in accordance with those in literature [23].

The mechanical properties of Titanium Ti6Al-4V parts manufactured by PAW-WAAM alloy are similar to those obtained by the traditional manufacturing processes of this alloy. The differences observed in the arrangement of the grains that form the microstructure do not translate into a difference in the mechanical properties of the material.

### 3.2. Torque Analysis during Cutting PAW-WAAM Walls

The torque signal has been recorded during each of the machining tests carried out by means of a Spike^®^ tool holder sensor. The torque signal leads to the understanding of the cutting effort necessary to carry out the chip removal. The following Figure 9 shows the evolution of the torque signal during the cut in up-milling and down-milling for the same cutting conditions with (ap = 5 mm, ae = 0.4 mm, Vc = 60 m/ min, fz = 0.084 mm/tooth). The signal shows a first phase in which the cutting edges are activated as the tool exceeds the outer limit of the piece with its entire diameter. An intermediate phase in which the signal has a constant regime from which the average torque value needed for the cut is extracted, and a last phase where the signal goes down to zero values with an abrupt slope when the process ends. Both signals show similar torque values for this set of conditions.

Figure 10 shows the comparison of the cutting torque for the different feed conditions per tooth between the up-milling and down-milling strategy. The signal shows the average values of the torque during its stable cutting phase for the two tests performed in each condition and the variance associated with that mean. It can be concluded that the cutting efforts are greater as we increase the feed per tooth, which is reasonable, since the cutting area increases. On the contrary, this increase in the torque signal is not linearly dependent, since there is a greater jump in the effort when it goes from 0.102 mm/tooth to 0.12 mm/tooth feed, while said increase is much smaller and can even be considered comparatively similar between feeds per tooth 0.089 and 0.102 mm/tooth.

Regarding the comparison of up-milling and down-milling strategies, the average values show higher values in case of up-milling, which seems to indicate that the down-milling strategy is slightly more favorable from the point of view of torque. This difference is not statistically significant given the variance observed in the analyzed sample, so considerations on the quality of the surface and other offered for the machining of these alloys manufactured by conventional technologies could be considered in this case [25].

Figure 11 shows the torque values during the cutting test at a cutting speed equal to 50 m/min and a speed of 60 m/min. It is observed that the torque is highly sensitive to the cutting speed used. The torque recorded at cutting speed 60 m/min is close to half that acquired during milling at a cutting speed of 50 m/min. This could be extrapolated from considerations made on machinability of Ti-6Al4V by Arrazola [26].

The average torque during milling at different depths corresponding to different Z positions of the tool when cutting are shown in Figure 12. The cutting torque at different depths is equivalent, so the slight increase in hardness in the middle cutting area does not have a significant effect.

### 3.3. Surface Quality

In this section, the surface quality results of the piece are analyzed in the different tests carried out based on the surface roughness measured in the PAW-WAAM mechanized wall.

Figure 13 shows the average roughness values at different feed rates per tooth in face milling in both up milling and down milling. In order to obtain the average roughness, three measurements of the profile of the two mechanized walls have been made first with each set of conditions, from each of which the arithmetic average of the absolute values of the roughness (Ra) has been extracted. The roughness values are typical of the finishing pass with an adequate magnitude. These values allow us to conclude that the quality of the piece is better when the up-milling strategy is used. Although down milling usually gives a better surface quality, in this case, the up-milling strategy led to a lower Ra value. This particular effect was also observed in a previous study carried out by Oliveira et al. [27]. They conclude that the origin of this result could be linked to the fact that the most vulnerable element in terms of vibration is the part and not the cutting tool. For down milling, a more periodical surface profile was observed, but the peak-to-valley height was higher than for up milling.

Regarding the roughness analysis in the passes made at different depths, the average value tends to be lower as it is machined at a greater depth, as can be seen in Figure 14. This could be related to the fact of having a more stable cut, with less deformation and vibrations, when the distance between the mooring of the piece and the cutting area where the tool acts is smaller. The Ra values measured even in the worst-case scenario are below 2 microns, which makes the wall present an optimum quality.

### 3.4. Tool Breakage Detection Based on Polar Analysis of Bending Moments

The evolution of torque during slot milling was also studied as part of the set of experiments. Table 2 shows the cutting conditions during milling across the wall. The cutting direction is perpendicular to the deposition direction of the weld that make up the wall, as it can be seen in Figure 15. The tool enters through the part of the sheet that serves as a 16 mm thick substrate. In carrying out the test when the tool was close to finishing the operation, there was an abrupt rupture of one of the cutting edges, which prevented its completion. The breakage may be conditioned by the fact that multiple interleaves of the weld beads must be cut along the path of the tool; in those faces, you can find phenomena of local hardening or small impurities that have ended up dramatically affecting the life of the tool.

The following Figure 16 shows the signals acquired by the Spike^®^ sensor during the slot milling test. The signals show the normal output at the end of the process due to the premature breakage of one of the cutting edges. The magnitudes that have been measured during the slot milling are the bending moment, torque, and the tension (force on the Z direction considering the axis shown on Figure 2).

Furthermore, the Spike^®^ measuring system registers the bending moment both on the X and Y axis over time. If the intersection of these bending moments is displayed in a 2D graph (as shown in Figure 17), the chip load on a cutter at a given point of time can be determined. This graph is known as a “polar plot”. If the mean value of the evolution of bending moment over the whole machining process is depicted in such a way, the condition of each cutting edge can be precisely defined. For example, this tool can be used to determine a non-uniform wear of the milling cutter.

## 4. Conclusions

In this work, the manufacturing of Ti6Al4V by a combined PAW-WAAM and milling processes has been studied. Based on the work done, the following conclusions can be drawn:-The mechanical properties of the wall manufactured by PAW-WAAM are in accordance with the aeronautical standards. No significant discrepancies have been observed in the mechanical properties at different positions and orientations of the specimens on the PAW-WAAM wall provided. The elastic limit, tensile strength, and elongation in the fracture values are above to those provided by the standard.-Regarding the effect of milling strategies on the mechanical stress of cutting and the surface quality of the final workpiece, up milling and down milling shows similar torque on average. Up milling shows slightly greater values of the torque, but the quality of the final surface is improved by this strategy compared to down-milling strategy.-On the analysis of different path and cutting speed considerations, although the torque does not seem to be sensitive to the depth in the Z direction of the workpiece, it seems highly sensitive to the cutting speed. The depth affects the surface quality improving as the tool is closer to the fixture system.-In the slot-milling test, a premature tool breakage was observed, and the effect of the tool path is responsible of this phenomenon. Direction perpendicular to the solidified weld metal is not convenient to the interface crossing of the tool. This premature breakage has been detected by the polar graph of the bending moment enabling the monitoring of the state of the milling edge.

## Figures and Tables

**Figure 1 materials-13-00766-f001:**
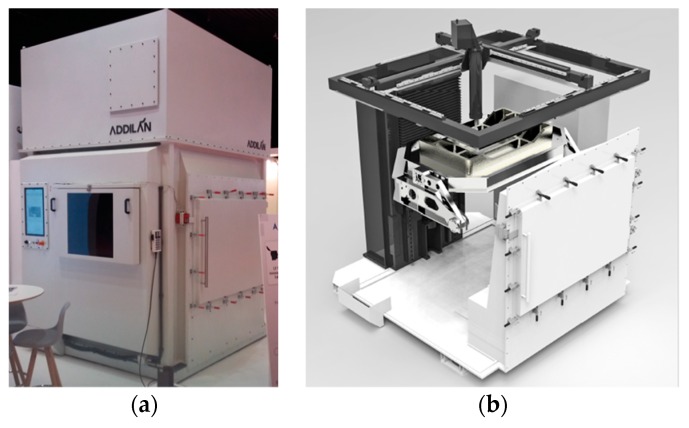
(**a**) General view of the machine; (**b**) Plasma equipment.

**Figure 2 materials-13-00766-f002:**
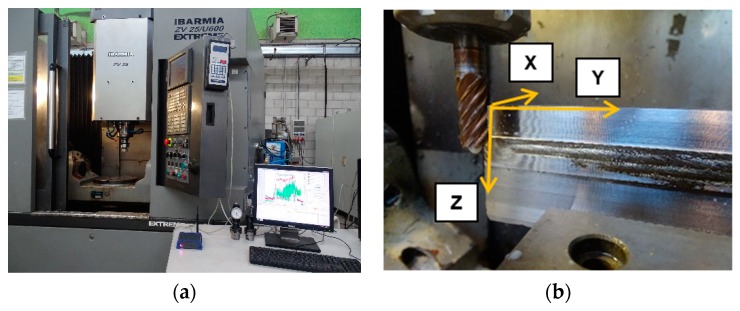
Machining experimental set-up: (**a**) machining equipment and (**b**) wall preparation.

**Figure 3 materials-13-00766-f003:**
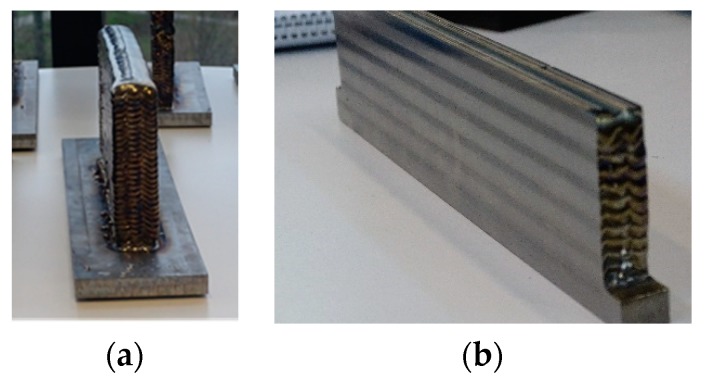
Walls preparation by Plasma Arc Welding (PAW)-Wire Arc Additive Manufacturing (WAAM) technologies: (**a**) wrought wall (**b**) machined wall.

**Figure 4 materials-13-00766-f004:**
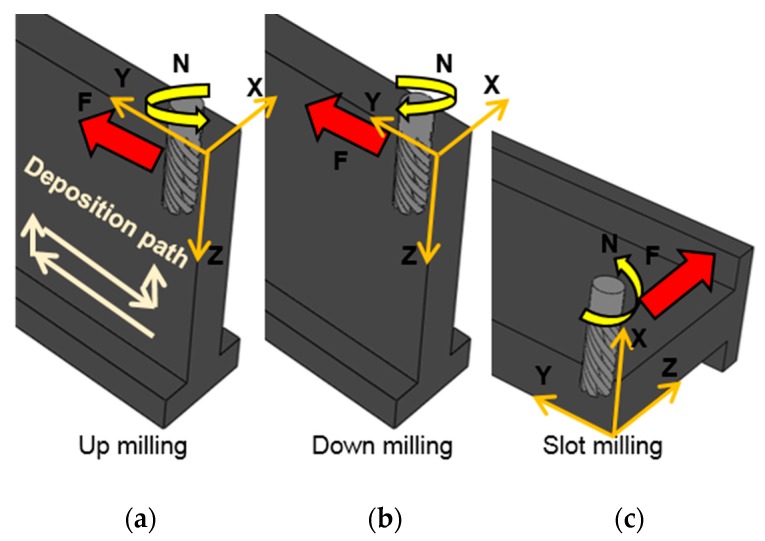
Tool path strategies for the face milling in different strategies (**a**) Up milling; (**b**) Down milling; (**c**) Slot milling.

**Figure 5 materials-13-00766-f005:**
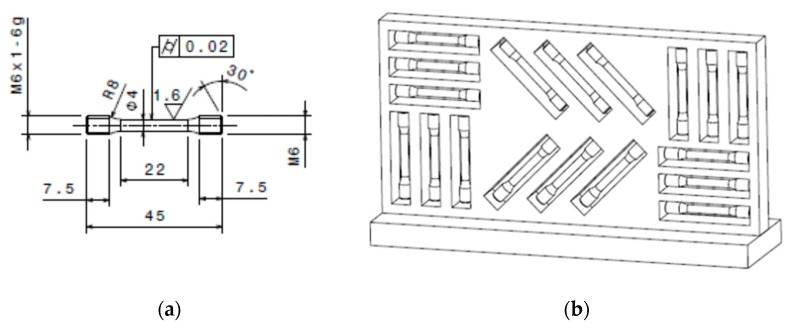
Specimens for tensile tests extracted from Ti6Al-4V PAW-WAAM walls: (**a**) Specimen drawing for evaluating the wall material following ISO 6892-1 standards; (**b**) Specimens in horizontal and vertical direction for evaluating the added material.

**Figure 6 materials-13-00766-f006:**
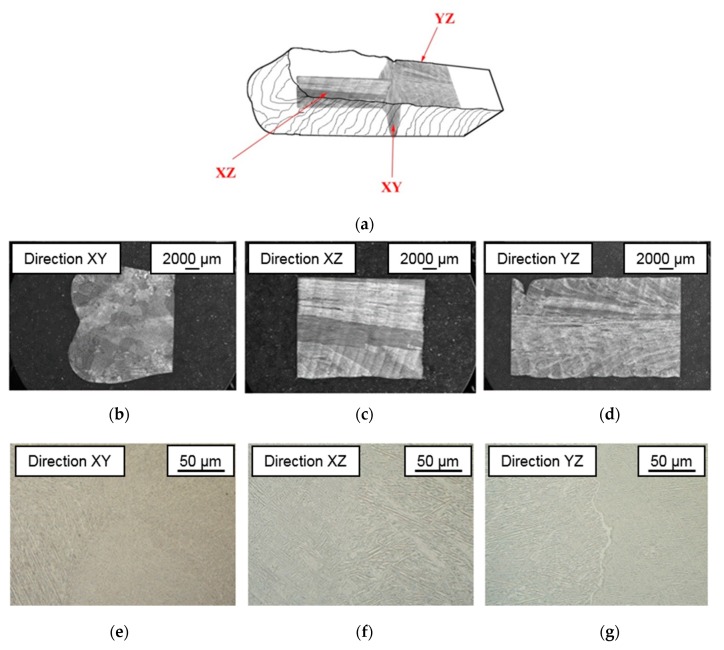
Macrographs and microstructure of the wall manufactured by PAW-WAAM in every direction. (**a**) General view of the planes; (**b**) macrograph on XY direction; (**c**) macrograph on XZ direction; (**d**) macrograph on YZ direction; (**e**) microstructure on XY direction; (**f**) microstructure on XZ direction and (**g**) microstructure on ZY direction.

**Figure 7 materials-13-00766-f007:**
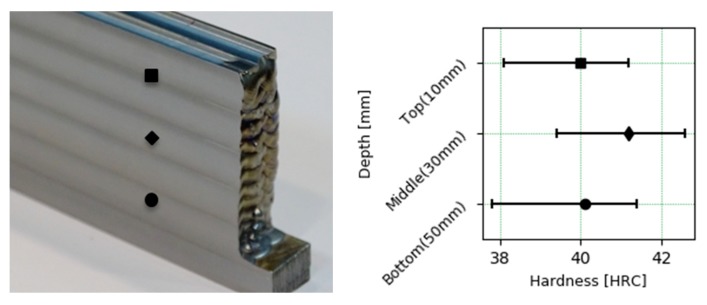
Hardness profiles of the walls.

**Figure 8 materials-13-00766-f008:**
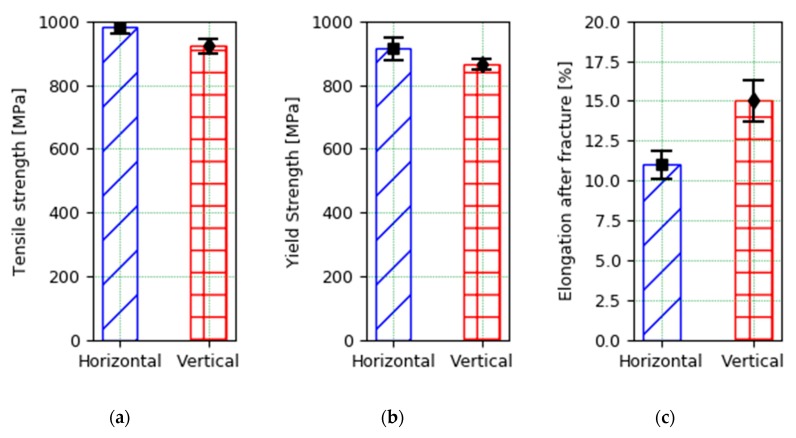
Mechanical properties of the Ti6Al4V PAW-WAAM wall. (**a**) Tensile strength; (**b**) Yield strength; (**c**) Elongation after fracture.

**Figure 9 materials-13-00766-f009:**
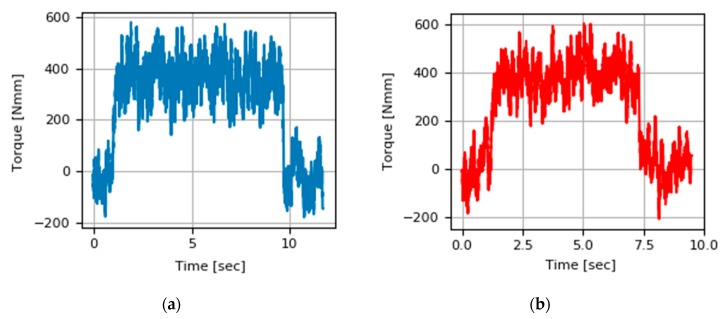
Torque signal during cutting PAW-WAAM wall with (ap = 5 mm, ae = 0.4 mm, Vc = 60 m/min, fz = 0.084 mm/tooth): (**a**) Up-milling strategy; (**b**) Down-milling strategy.

**Figure 10 materials-13-00766-f010:**
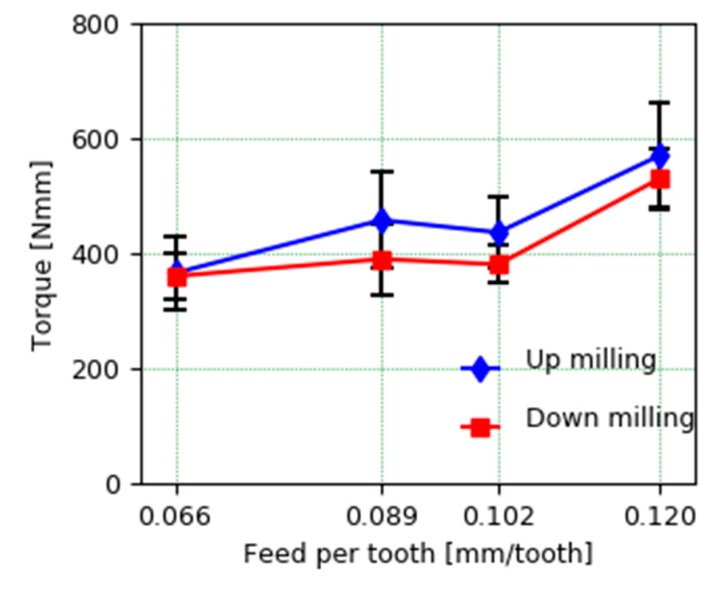
Comparison of mean torque at different cutting strategies and feed per tooth.

**Figure 11 materials-13-00766-f011:**
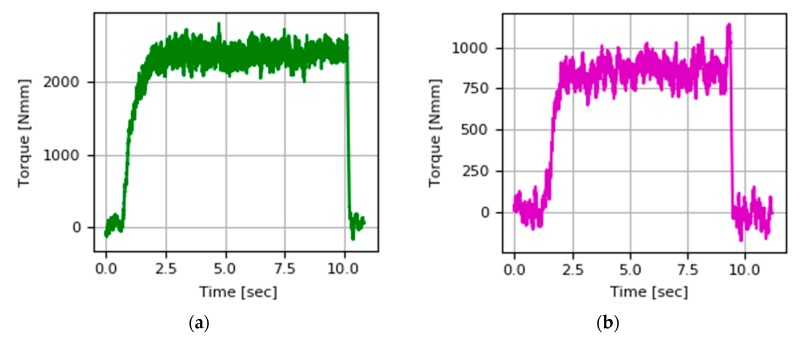
Torque signal during cutting PAW-WAAM wall with (ap = 0.5 mm, ae = 0.4 mm, fz = 0.12 mm/tooth) at different cutting velocities: (**a**) Vc = 50 m/min, (**b**) Vc = 60 m/min.

**Figure 12 materials-13-00766-f012:**
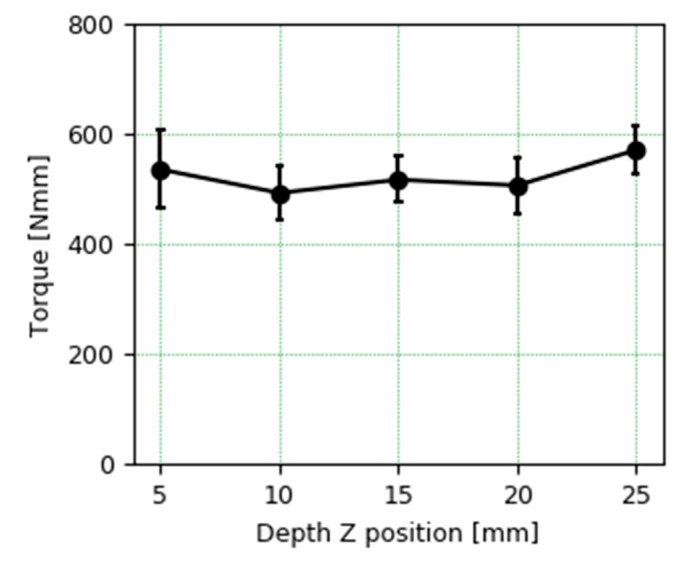
Torque comparison on cutting PAW-WAAM wall at different Z tool positions from 5 mm to 25 mm.

**Figure 13 materials-13-00766-f013:**
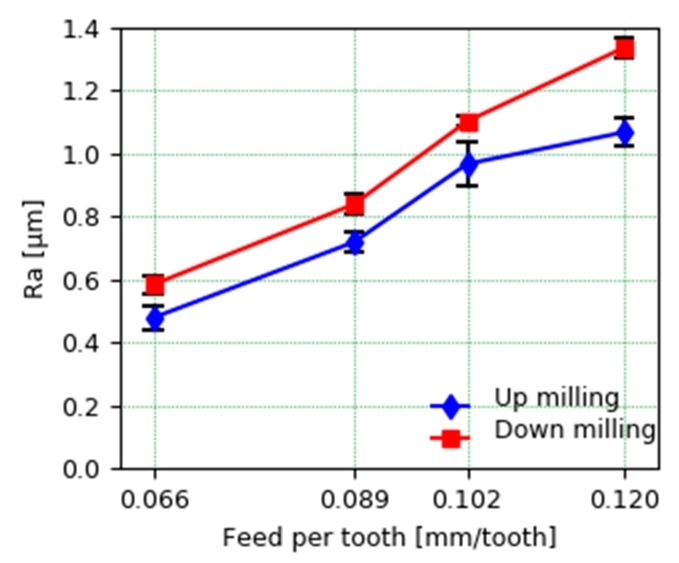
Average roughness (Ra) at different cutting strategies and feed per tooth.

**Figure 14 materials-13-00766-f014:**
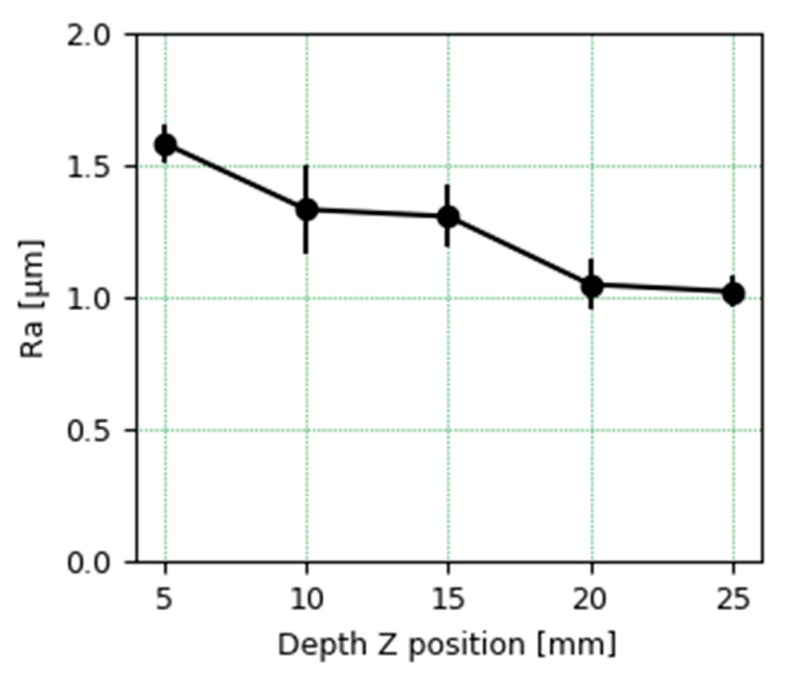
Average roughness (Ra) at different Z tool positions from 5 mm to 25 mm.

**Figure 15 materials-13-00766-f015:**
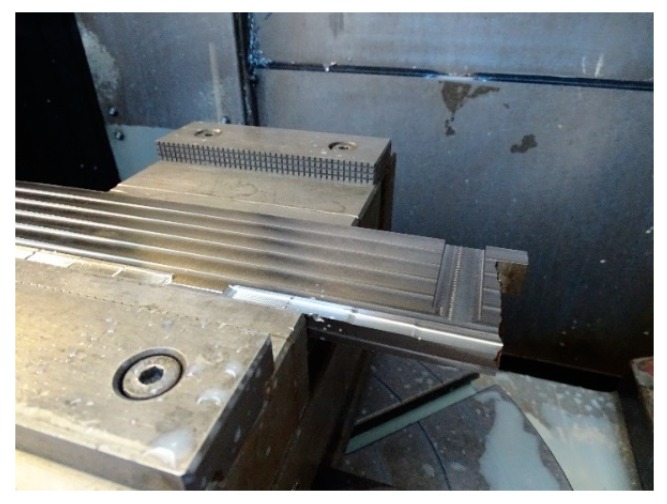
Slot milling across the PAW-WAAM wall.

**Figure 16 materials-13-00766-f016:**
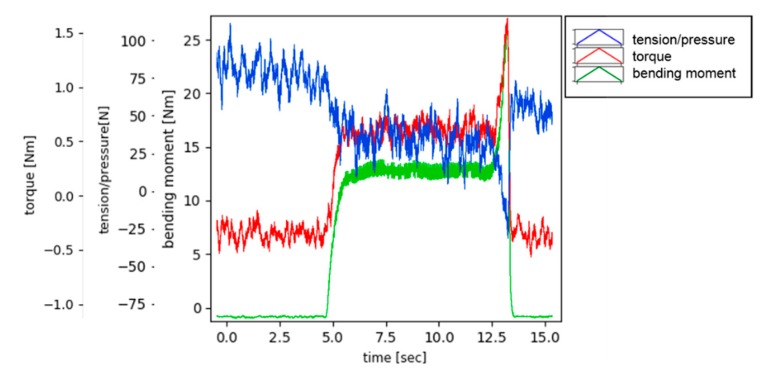
Signals acquired during slot milling across the wall.

**Figure 17 materials-13-00766-f017:**
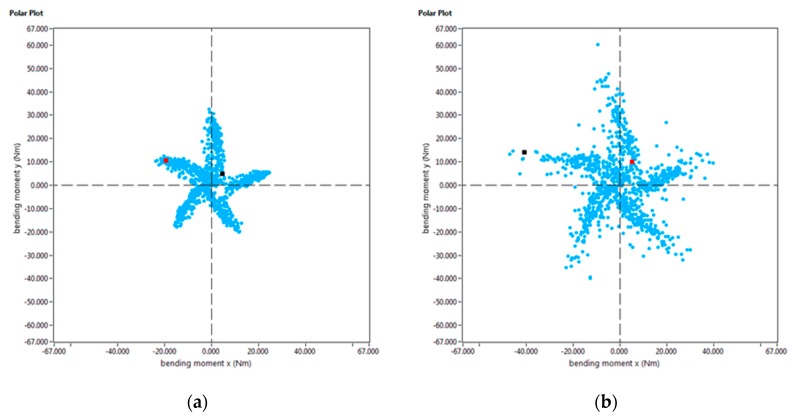
Polar plot of the average bending moment in the slot milling experiments: (**a**) normal behavior of the bending moment; (**b**) Polar diagram during the breakage of the teeth.

**Table 1 materials-13-00766-t001:** Chemical composition of the wire material employed (wt %).

C	O	N	H	Fe	Al	V
0.02	0.16	0.011	0.0037	0.07	6.19	3.92

**Table 2 materials-13-00766-t002:** Cutting conditions in the feed speed and milling strategy tests.

Milling Strategy	ap	ae	fz	Vc
	mm	mm	mm/tooth	m/min
Up-milling/Down-milling	5	0.4	0.066	60
			0.084	
			0.102	
			0.12	
Slot milling	0.4	12	0.066	60

**Table 3 materials-13-00766-t003:** Cutting conditions in the analysis of the influence of the depth (Z position of the tool).

Milling Strategy	ap	Ae	fz	Vc	Depth (Z Position)
	mm	Mm	mm/tooth	m/min	Mm
Up-milling	5	0.4	0.12	60	5
					10
					15
					20

**Table 4 materials-13-00766-t004:** Cutting conditions in the cutting speed tests.

Milling Strategy	Ap	ae	fz	Vc
	Mm	mm	mm/tooth	m/min
Up-milling	5	0.4	0.12	50
Up-milling	5	0.4	0.12	60

**Table 5 materials-13-00766-t005:** Chemical composition of the PAW-WAAM material compared to the AMS4928 standard (wt %) related to forged material.

Chemical Element	Ti6Al-4V		Ti6Al-4V PAW-WAAM
	min.	max.	
C	0	0.08	0.02
O_2_	0	0.25	0.16
N_2_	0	0.03	0.014
H_2_	0	0.0125	0.003
Fe	0	0.3	0.17
Al	5.5	6.75	6.3
V	3.5	4.5	4

**Table 6 materials-13-00766-t006:** Results of tensile testing of PAW-WAAM specimens. UTS: ultimate tensile strength, YS: yield stress.

Sample	UTS [MPa]	YS [MPa]	Elong [%]
Horizontal PAW-WAAM (PTH)	981 ± 36	917 ± 19	11 ± 0.9
Vertical PAW-WAAM (PTV)	925 ± 18	864 ± 22	15 ± 1.3
Ti6Al4V (AMS4928)	931	862	10

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
