# Peer review of "Analysis of the Machining Process of Titanium Ti6Al-4V Parts Manufactured by Wire Arc Additive Manufacturing (WAAM)"

_materials, 2020, doi:10.3390/ma13030766_

Round 1
Reviewer 1 Report
The paper presents the detailed analysis of machining process of machine parts manufactured by Wire Arc Additive Technology. The article may particularly interesting to researchers involved in machining titanium alloys. In general, it should be emphasized that extensive experimental research has been carried out, however the paper requires some revisions. First of all, the manuscript requires many language corrections and should be carefully checked in this terms. The review pointed to several language errors, of which there are certainly more.
The detailed remarks are as follows:
Line 38: Instead of “(…) manufacture pieces” I recommend using “manufacture machine parts” or “(…) workpieces”
Line 87: It would be better to replace “(…) characteristics” with ex. “features”
Line 89: “produce the walls” does not sound good.
Lines 90-92: The sentence “A set of two walls (…)” should be rewritten. It can be concluded that machining of the second wall was carried out in order to carry out machining tests, which makes no sense.
Line 93: there is no space between “1.2mm” and “wire” and the description in brackets should be explained.
Line 105: When writing about the axis directions, they should be marked on the drawing, or at least explained.
Line 110-111: The sentence “The diameter tool (…)” should be rewritten because it is not linguistically correct.
Line 111: The tool material cannot be seen in the figure.
Line 119: What you mean by “clearing process”?
Line 119-120: The sentence “It can be observed(…) “ should be rewritten – it is not understood.
Line 121: “planification” is not a good word in this context – it should be changed.
Figure 4 and description of the tool strategy: Figure colors should be changed, because the arrows and description of tool path are almost invisible. Are the arrows for upmilling path correct? How is it possible to carry out machining starting from the base of the part?
Line 136 – 137 and Table 3: Since the cutting depth is ap, the “depth (z position) is misleading. My guess is that this is related to the number of tool passes, and "Z position" is the product of the ap depth and the number of passes, but it should be clearly explained.
Lines 149-151: The aim of the experiment should be clearly stated at the beginning of the description and not at the end. Furthermore, the sentence is linguistically incorrect (ex. too many “and”) and should be rewritten.
Figure 6: The figure contains 7 different drawings that are not signed a), b), c) ... etc.
Figures 9-12: Since these drawings show the same measured physical quantity (torque), why are there different names in the signatures? Moreover, in some cases there are erroneous units.
Lines 256 – 258: The authors state that the surface quality for up milling is better than for down milling. However, down milling usually gives a better surface quality, which I can tell from the practice of machining steel alloys and what is often given in milling theory. If the machining of titanium alloys gives different results, neglecting commonly known phenomenon, I consider it necessary to explain why it did. Moreover, please indicate the paragraph in which the authors of the work [27] obtained the same results.
Figures 13-14, line 265: “micron” should be replaced by the correct form of the length unit “micrometre”.
Figure 16: The figure is illegible. Should be changed and additionally explained in the text.
Author Response
We would like to thaks to the reviewer for their detailed comments and suggestions for the manuscript. Below, you will find a point by point description of how each comment was addressed in the manuscript. Original reviewer comments in boldface, responses in regular typeface.
The paper presents the detailed analysis of machining process of machine parts manufactured by Wire Arc Additive Technology. The article may particularly interesting to researchers involved in machining titanium alloys. In general, it should be emphasized that extensive experimental research has been carried out, however the paper requires some revisions. First of all, the manuscript requires many language corrections and should be carefully checked in this terms. The review pointed to several language errors, of which there are certainly more.
Response: We would like to thank to the reviewer for their comments and suggestions for the manuscript.
The detailed remarks are as follows:
Line 38: Instead of “(…) manufacture pieces” I recommend using “manufacture machine parts” or “(…) workpieces”
“Therefore, WAAM is the best solution to manufacture machine parts…”
Line 87: It would be better to replace “(…) characteristics” with ex. “features”
“One of the main features of this equipment is its specific…”
Line 89: “produce the walls” does not sound good.
“This machine was used to manufacture the walls of…”
Lines 90-92: The sentence “A set of two walls (…)” should be rewritten. It can be concluded that machining of the second wall was carried out in order to carry out machining tests, which makes no sense.
“The wall was separated into a set of two samples machined on a CNC machining…”
Line 93: there is no space between “1.2mm” and “wire” and the description in brackets should be explained.
“was employed and 1.2 mm wire diameter (AWS A5.16-13 ERTi-5 Titanium Welding Wire that meet standard AMS-4954J)”
Line 105: When writing about the axis directions, they should be marked on the drawing, or at least explained.
Authors add the axis on the Figure 2b as follows:
Line 110-111: The sentence “The diameter tool (…)” should be rewritten because it is not linguistically correct.
“Tool diameter and teeth number of the KENDU milling tool are 12 mm and 5 teeth…”
Line 111: The tool material cannot be seen in the figure.
“The milling tool, which it can be seen in Figure 2b, is made of WC-8%Co material”
Line 119: What you mean by “clearing process”? Line 119-120: The sentence “It can be observed(…) “ should be rewritten – it is not understood
It has been changed to:
“Figure 3b illustrates the machined wall after preparation, machining from wrought material to semi-finished material, of the surface to performed cutting tests and measurement of the material properties.”
Line 121: “planification” is not a good word in this context – it should be changed.
“Before explaining the tests to be performed”
Figure 4 and description of the tool strategy: Figure colors should be changed, because the arrows and description of tool path are almost invisible. Are the arrows for upmilling path correct? How is it possible to carry out machining starting from the base of the part?
Figure 4 has been changed to make it more visible that the path shown is the deposition path not the machining tool path. What made the interpretation of the figure misleading for the readers.
Line 136 – 137 and Table 3: Since the cutting depth is ap, the “depth (z position) is misleading. My guess is that this is related to the number of tool passes, and "Z position" is the product of the ap depth and the number of passes, but it should be clearly explained.
Yes, you are right we hope to clarify it with this change:
“As mentioned before, Table 3 has an additional parameter called depth. The objective is to study the influence of Z position of the tool, which is the product of the ap and the pass starting from the top of the wall, when Up-milling operation is been performing with constant cutting parameters.”
Lines 149-151: The aim of the experiment should be clearly stated at the beginning of the description and not at the end. Furthermore, the sentence is linguistically incorrect (ex. too many “and”) and should be rewritten.
Finally, the cutting parameters used to analyse the effect of changing the cutting velocity when milling a PAW-WAAM part are shown on Table 4.
Figure 6: The figure contains 7 different drawings that are not signed a), b), c) ... etc.
|
(a) |
||
|
(b) |
(c) |
(d) |
|
(e) |
(f) |
(g) |
Figures 9-12: Since these drawings show the same measured physical quantity (torque), why are there different names in the signatures? Moreover, in some cases there are erroneous units.
The authors regret having made that coherence error in the way of referring to the torque and thank the reviewer for the comment. The figures have been modified and unified
Lines 256 – 258: The authors state that the surface quality for up milling is better than for down milling. However, down milling usually gives a better surface quality, which I can tell from the practice of machining steel alloys and what is often given in milling theory. If the machining of titanium alloys gives different results, neglecting commonly known phenomenon, I consider it necessary to explain why it did. Moreover, please indicate the paragraph in which the authors of the work [27] obtained the same results.
Thank you for this remark. The paragraph has been modified to explain this result:
Despite down milling usually gives a better surface quality, in this case, the up milling strategy led to a lower Ra value. This particular effect was also observed in a previous study carried out by Oliveira et al. [ref]. They conclude that the origin of this result could be linked to the fact that the most vulnerable element in terms of vibration is the part and not the cutting tool. For down milling a more periodical surface profile was observed, but peak to valley height was higher than for up milling.
Figures 13-14, line 265: “micron” should be replaced by the correct form of the length unit “micrometre”.
We have chosen the Greek symbol: µ.
Figure 16: The figure is illegible. Should be changed and additionally explained in the text.
We have tried to improve the figure by making it more legible. The text: “The magnitudes that have been measured during the slot milling are the bending moment, torque and the tension (force on the Z direction considering the axis shown on Figure 2).”

Reviewer 2 Report
Table 2. The presented values can be misunderstood by readers - please add the horizontal line to separate parameters of Up-/Down-milling and Slot milling
Fig. 6. Macro- and microstructures are unreadable - please change in the photo with better quality, esp. the scale and orientation directions must be readable.
To consider (acc. to Fig 8 and table 6): In my opinion the results of UTS, YS and Elongation is better presented as the lowest value or as the range of results. What it means 981+/-36.3, and how important is 0.3 MPa?
Author Response
Response to Reviewer 2 Comments
We would like to thank to the reviewer for their detailed comments and suggestions for the manuscript. Below, you will find a point by point description of how each comment was addressed in the manuscript. Original reviewer comments in boldface, responses in regular typeface.
Table 2. The presented values can be misunderstood by readers - please add the horizontal line to separate parameters of Up-/Down-milling and Slot milling
We have changed Table 2 as follows
Table 2. Cutting conditions in the feed speed and milling strategy tests
|
Milling process |
ap |
ae |
fz |
Vc |
|
mm |
mm |
mm/tooth |
m/min |
|
|
Up-milling/Down-milling |
5 |
0,4 |
0,066 |
60 |
|
0,084 |
||||
|
0,102 |
||||
|
0,12 |
||||
|
Slot milling |
0,4 |
12 |
0.066 |
60 |
Fig. 6. Macro- and microstructures are unreadable - please change in the photo with better quality, esp. the scale and orientation directions must be readable.
We try to better show the microstructure giving more quality to the image and having more readable labels.
|
(a) |
||
|
(b) |
(c) |
(d) |
|
(e) |
(f) |
(g) |
To consider (acc. to Fig 8 and table 6): In my opinion the results of UTS, YS and Elongation is better presented as the lowest value or as the range of results. What it means 981+/-36.3, and how important is 0.3 MPa?
Authors have changed the resolution of the value but since it is that close to the meeting of the standard we want to address that the result is not always bellow the value. It may be necessary to carry out some subsequent treatment to ensure that we are in values greater than the minimum required by the aeronautical industry
|
|
UTS [MPa] |
YS [MPa] |
Elong [%] |
|
Horizontal PAW-WAAM (PTH) |
981±36 |
917±19 |
11±0.9 |
|
Vertical PAW-WAAM (PTV) |
925±18 |
864±22 |
15±1.3 |
|
Ti6Al4V (AMS4928) |
931 |
862 |
10 |

Reviewer 3 Report
In this study, the manufacturing of Ti6Al4V with a PAW-WAAM and milling process studies has been investigated, which includes compositions, microstructure characterization, Mechanical properties, Torque analysis and Surface quality analysis. In general, this manuscript is in good shape. It is deserved to be published after major revision.
1. The "Discussion" is missing. For example, as shown in Table 6, the UTS, YS and elongation of AM parts are better than the reference, why?
Author Response
We would like to thank to the reviewer for their detailed comments and suggestions for the manuscript. Below, you will find a point by point description of how each comment was addressed in the manuscript. Original reviewer comments in boldface, responses in regular typeface.
In this study, the manufacturing of Ti6Al4V with a PAW-WAAM and milling process studies has been investigated, which includes compositions, microstructure characterization, Mechanical properties, Torque analysis and Surface quality analysis. In general, this manuscript is in good shape. It is deserved to be published after major revision.
The "Discussion" is missing. For example, as shown in Table 6, the UTS, YS and elongation of AM parts are better than the reference, why?The results shown do not refer to the comparison with the material from another type of manufacturing, but to the requirements of the aeronautical standard, which in this case is satisfied. An indicative text has been added addressing this issue. Also, a discussion on the surface quality on both strategies has been included to make the impact of this paper more interesting.

Reviewer 4 Report
In this manuscript, the authors have conducted the analysis of the Machining Process of Titanium Ti6Al-4V Parts Manufactured by Wire Arc Additive Manufacturing (WAAM). The effects of mining strategies and feed, the Z depth and the cutting velocity on the torque and surface quality of the final PAW-WAAM part. The English is readable and the arrangement of this manuscript is reasonable. The reviewer would like to recommend this manuscript as major revision.
Author Response
We would like to thank the reviewer for their detailed comments and suggestions for the manuscript. Below, you will find a point by point description of how each comment was addressed in the manuscript. Original reviewer comments in boldface, responses in regular typeface.
In this manuscript, the authors have conducted the analysis of the Machining Process of Titanium Ti6Al-4V Parts Manufactured by Wire Arc Additive Manufacturing (WAAM). The effects of mining strategies and feed, the Z depth and the cutting velocity on the torque and surface quality of the final PAW-WAAM part. The English is readable and the arrangement of this manuscript is reasonable. The reviewer would like to recommend this manuscript as major revision.
We have done some changes to the paper trying to meet your expectations and willing to have good scientific quality.

Round 2
Reviewer 3 Report
It is ready for publishing after correcting one typo.
In the right graph of Fig. 7, "Midle" should be "Middle".
Author Response
We would like to thank the reviewer for their detailed comments and suggestions for the manuscript. Below, you will find a point by point description of how each comment was addressed in the manuscript. Original reviewer comments in boldface, responses in regular typeface.
It is ready for publishing after correcting one typo.
In the right graph of Fig. 7, "Midle" should be "Middle".
We have done the change on the paper, we apologize for the error on tipping and thanks again for the review.
